# On the legal nature of synthetic data

**César Augusto Fontanillo López**[1*]     **Abdullah Elbi**[1*]
[1]Center for IT and IP Law
KU Leuven
Leuven, 3000, Belgium
`cesar.fontanillo@kuleuven.be`

## Abstract

The present manuscript attempts to analyse the legal qualification of synthetic data generated from personal data. Three main conclusions are drawn from our legal analysis: first, full data protection compliance prior to data synthesis would be applicable in many cases; second, according to the identifiability test as enshrined in the definition of personal data, synthetic data will be considered pseudonymous or anonymous data depending on the appropriateness of the data synthesis and the related ex-post control mechanisms; third, the broader question of the legal qualification of synthetic remains an unresolved issue in light of the exegetical discrepancy by law and doctrine on the identifiability test and regulatory model for data protection law.

## 1   Disclaimer

Synthetic data is a broad concept encompassing both personally and non-personally identifiable information. This manuscript focuses, notwithstanding, on the intersection between synthetic data and personal data. The reasons for so doing are twofold. First, generating synthetic data by means of personal data (including hybrid data [1]) simplifies our scope of analysis because we avoid entering into the lively academic debate [2] on the concept of personal data prior to the study of the legal qualification of synthetic data. This relaxes the conditions of our assessment, as the qualification of existing models and background knowledge used as sources for data synthesis is problematic [3]. Second, the consideration of personal data as a starting point allows us, consequently, to provide a more straightforward assessment of the legal qualification of synthetic data because we can depart from a given premise on which to anchor and study said class, thus considerably reducing the degrees of freedom of our inquiry.

## 2   Introduction

Synthetic data is attracting increasing attention from technicians [4] and legal scholars[5] [6] in recent years. This is especially noticeable among entities and people working on data-driven technologies, particularly in the artificial intelligence application development and testing sector, where sheer volumes of data are needed. In these circles, synthetic data has become a growing trend by promising to alleviate existing data access and analytics challenges while respecting data protection rules. Given the rising prospects and acceptance of data synthesis, there is a need to assess the legal implications of its generation and use, the starting point being the legal qualification of synthetic data.

In the present manuscript, we attempt to study the legal qualification of synthetic data generated by means of personal data in accordance with the European data protection framework. For this purpose, we focus our analysis on synthetic data as anonymous and pseudonymous data in relation to the identifiability threshold as set out in the General Data Protection Regulation (GDPR) [7]. Prior to

NeurIPS 2022 Workshop on Synthetic Data for Empowering ML Research.

this analysis, we briefly introduce the concept of synthetic data with the aim of contextualizing the legal debate on identifiability.

# 3 Definition

Synthetic data has been named in varying ways, such as 1fake data' [8] or 'artificial data' [9]. Regardless of the terminology, synthetic data is, at a fundamental level, data artificially generated from original data that preserves the statistical properties of said original data [4]. This artificial process of data generation is normally performed by a machine learning model, which captures the structure and statistical distribution of original data to produce a synthetic data. The conservation of the statistical properties of the original data in the synthetic data is crucial, as it allows data analysts to draw meaningful conclusions from the synthetic data as if they were drawn from the original data [4].

During the synthetic data generation process, or data synthesis, a certain degree of randomness can be induced in the synthetic data, unrelated to the original data, to produce data sets with high variability [10]. This allows the creation of extensive data sets of heterogeneous characteristics that can be used for a multiplicity of purposes. At the same time, the level of randomness in the data generation process can be controlled to ensure that the synthetic data is sufficiently diverse, yet still concordant with the original data [10].

# 4 Preliminary considerations of the legal nature of synthetic data

The establishment of the legal nature of synthetic data is a complex task comprising the validation of different elements. Some of them are easily assessable, while others are difficult to identify and determine. The ease or difficulty in assessing these elements is conditioned by their degree of interpretability or, in other words, by the level of consistency between the legal qualification made by an individual or entity with respect to the synthetic data and the legal qualification agreed upon by the legal system. If an individual or entity can consistently predict the juridical response of the legal system with respect to the envisaged qualification, the element will be easily validated, and conversely if not.

The degree of assessability can be influenced by the notion of risk inherent to the European data protection framework and, more particularly, to the concept of personal data, a legal construct for which the risk of re-identification plays a fundamental role in the establishment of its legal nature. Because risk is likely to evolve over time depending on the context, events, time, or agents, [11] the very nature of risk affects the legal determination of synthetic data in the way that, under certain circumstances, synthetic data will be considered anonymous data, while in others, this will not be the case. The consideration of risk is, henceforth, an important element in the analysis of synthetic data, and one that prevents the possibility of legally qualifying synthetic data in an absolute manner. The assessment of the legal nature of synthetic data must be therefore carried out contextually, in attention to the changing nature of risk, so that the legal nature of synthetic data can be sustained or transmuted over time, depending on the circumstances.

# 5 Synthetic data as personal data

Any legal qualification of synthetic data from the perspective of European data protection law must depart from the definition of personal data as it is the dichotomy between personal and non-personal data that synthetic data must navigate. According to Article 4(1) GDPR, personal data means:

> *'any information relating to an identified or identifiable natural person.'*

**I**n validating the elements of the provided definition, recourse to the degree of assessability as previously addressed will be made. In our view, two groups of elements can be differentiated with respect to their level of assessability. On the one hand, we consider the elements 'any information' and 'natural person' as easily assessable classes. On the other hand, we consider the elements 'relating to' and 'identified or identifiable' as difficulty assessable classes.

## 5.1 Easily assessable classes

### 5.1.1 'Any information'

To begin with, synthetic data can be arguably categorized as 'any information' in all circumstances. This is so because the very essence of synthetic data is informational, irrespective of its nature, content, or format [12]. In other words, synthetic data can be categorized as 'any information' regardless of whether synthetic data is accurate or inaccurate (nature), objective or subjective (content), or generated by certain types of machine learning models or others (format).

One must note that information seems to be treated by European data protection law as meaning the same as data, *i.e.* it treats both concepts interchangeably. This is an important analytical consideration since, by so doing, the European data protection framework takes an all-encompassing approach to the qualification of information. Conventionally, information differentiates from data in the way that the former assumes the latter. It encompasses data (this is, a reinterpretable representation of information in a formalized manner suitable for communication, interpretation, or processings [13]) and the knowledge gained thereof through its analysis and interpretation within a certain context [14]. By equating information to data, the context under which personal data is observed becomes irrelevant. In other words, for European data protection law it is arguably of no concern whether personal data can constitute information for certain parties and under certain circumstances and not for others and other circumstances as long as data is being processed.

This particular regulatory choice obeys the fundamental rights perspective of European data protection law, which, together with ensuring the free flow of information, has also as a purpose the attainment of a harmonized level of protection of the fundamental rights of the data subject [15]. As a result of this inclination of the legislator towards the protection of fundamental rights, data becomes regulated from an 'absolute' perspective, regardless of the context in which it is ascribed. This ensures the enjoyment of legal protection of data in any situation, as long as personal and processed. If the opposite was pursued, namely the regulation of 'information' instead of 'data', the protection of the fundamental rights of the data subject would be relegated to the indetermination, as it would be hardly assessable whether a piece of data is considered information or not for a specific entity in a specific time given the difficulty of determining the informational sources that said entity can have [16].

At the same time, such an undertaking could not ensure an effective regulatory response to the nature of information, for which mutability is a fundamental intrinsic element. By using information as the object of regulation, the legal framework would be condemned to relativity, and therefore would not be fit for creating legal certainty. This would lead to an under-protection of fundamental rights, as data protection law would only be triggered in those cases in which objective knowledge of the informational sources of a given entity is established.

In order to avoid entering into the marshy ground of establishing when information is meaningfully extracted from data, it is more effective from a regulatory viewpoint to have data itself be the gravitational point of regulation, or, as the regulator does, to treat data and information as interchangeable concepts. Because information is equated to data under the European data protection framework, legal certainty is enhanced in the way that the regulator does not need to prove whether certain data constitutes information or not: that is an *iuris et de iure* presumption, or a conclusive presumption of law which cannot be rebutted by evidence. As a result, the qualification of the element 'any information' constitutes an assessment with a low degree of interpretability. The regulatee has the certainty that the synthetic data that is being processed qualifies as 'any information' within the remit of European data protection law.

### 5.1.2 'Natural person'

Synthetic data must refer to the 'natural person' in order to be considered personal data. The natural person is also an easily assessable legal construct. It is referred to in Article 6 of the Universal Declaration of Human Rights and its definition can be normally found in national civil law [12]. In many cases, the natural person refers to the living individual considered as a subject of rights and obligations [17].

Where synthetic data refers to the natural person, it will be considered personal data. For the purposes of our legal analysis, synthetic data generated from personal data will always concern the natural person within the established constraints of this paper, as personal data is the substrate upon which

synthetic data is generated. If this was not the case, for instance, because the original data would refer to deceased individuals or to other entities different from the natural person, such as inanimate objects, then it would 'generally' fall out of the scope of this analysis, as data protection law would not be applicable. The use of the term 'generally' is not trivial. In certain circumstances, such as for the processing of genetic data, there are voices advancing the qualification of data concerning deceased individuals as personal data for their biological family members [18], but also in other circumstances, such as where data of inanimate objects relates to an identified or identifiable natural person, personal data processing could be in place [2]. In addition to this, one must note that synthetic data could still be considered non-personal data outside the boundaries of this manuscript.

Closely related to the assessment of the concurrence of the 'natural person', we acknowledge that the validation of this element may be resource-consuming both in terms of time and cost, depending on the circumstances. For instance, we conceive it as onerous and tedious the activity of verifying whether the individuals contained in a data set are alive or not. However, the laboriousness of auditing should not be confused with the complexity of interpreting, for the former assumes a mechanical effort and the latter assumes a reflexive effort. When it comes to the assessability of this element in the terms described above, we are of the opinion that its determination will most likely not cause interpretative issues. It is highly foreseeable that, as long as it is determined whether an individual contained in a dataset is alive or not, the legal qualification carried out by the regulatee will equate to that made the legal data protection system.

## 5.2 Difficultly assessable classes

### 5.2.1 'Relating to'

As opposed to the previous elements, which enjoy high foreseeability standards, the elements 'relating to' and 'identified or identifiable' encounter a lower degree of predictability.

In the case of 'relating to', its validation merits detailed analysis. On a general note, 'relatedness' can be instantiated if a veridical attribution is established between entities under observation. This is a rational process supported by factual evidence that, despite its eventual mutability over time, can be causally established and verified with respect to said entities in a given moment of time. This would imply that, as long as a veridical attribution between entities at a given moment is found, those entities would relate to each other. Such exercise may not in principle appear as problematic. However, when one considers the dimension on which relatedness is to be assessed, problems start to appear.

Since relatedness is a relative property, or a property that has to be specified in comparison with something else, epistemological uncertainty arises as to the suitable dimension in which said comparison should be established. For data protection law purposes, this means, most obviously, uncertainty as to the subjective dimension from which to categorize a specific personal data processing. It is not the same, and certainly the legal effects will not be the same, if the evaluation of relatedness is conducted at microscale, considering the epistemological constraints of a given processing entity, than if it is conducted at macroscale, without considering any epistemological constraint. For instance, if a given entity possesses certain pieces of information that relate to an individual whose personal data is not held by such entity, doubt can be cast on whether one should consider those pieces of information as 'relating to' said individual or not. Depending on the regulatory policy adopted, diametrically opposed legal responses will be possible.

In addition to this dimensional problem, there are other practical considerations that must be tied up with the concept of relatedness. They most obviously refer to the building process of the concept of personal data and the shared attributes of its constituent elements. The intricacies of these particular considerations will be more extensively examined below, while dealing with the element 'identified or identifiable'. For the moment, it is sufficient to acknowledge that the element of relatedness will be conditioned, in most cases, by the element 'identified or identifiable'. In other words, the outcomes of the determination of relatedness would equate, in practical terms, to those obtained in the assessment of the 'identified or identifiable' natural person. The reason for so thinking is that the relative dimension of relatedness is already shared by and considered in the assessment of the 'identified or identifiable' natural person, as will be explained, so that the building process of personal data becomes a consistent undertaking with a clear regulatory direction. As long as the 'identified or identifiable' individual is determined, relatedness will follow suit.

While such an assumption may entail circular reasoning, and surely will deprive the element of relatedness of fully acquiring an autonomous meaning in European data protection law, different from the 'identified or identifiable', at least in its subjective dimension, reasonable regulatory objectives, such as the attainment of a harmonized interpretation of the concept of personal data, support this view. In fact, if a different stance would be advocated, *e.g.* that of having different regulatory approaches to the dimensional space of 'relating to' and 'identified or identifiable', epistemological inconsistency would emerge in the treatment of the constituent elements of personal data. This may imply contradictory regulatory views in the very concept of personal data, which would render it an unstable concept. Concomitantly, this would undermine an important *telos* of European data protection law, *e.g.* providing a 'harmonized' level of protection for personal data.

We would argue, to this extent, that information about an 'identified or identifiable' individual must 'always' relate to him or her within the remit of data protection law. An instance in which information relates to a non-identified or non-identifiable individual may indeed be possible, but as long as identifiability is not triggered, the consideration of said information as personal data would be at odds with the very idea of regulatory consistency, and should therefore be abandoned in data protection law discourse. Consequently, information relating to a non-identified or non-identifiable natural person would not merit the consideration of personal data, even if, theoretically, a veridical attribution could be established. Take, for instance, the notions of anonymous or aggregate data. In theory, relatedness could still be established for anonymous and aggregate data, *e.g.* if one knows that an individual is in a specific data set, but cannot individualize it, nor link or infer information about it, relatedness would still be present, but identifiability would not be triggered. In those cases, as long as personal data enters into this domain, it is, and should be, as we argue, in principle, out of the scope of the European data protection framework, irrespective of whether said data could theoretically relate to an individual.

Leaving aside this digression, further accentuation can be put on the ways in which information relates to the individual. According to the authoritative understanding of relatedness, information can relate to the natural person in content, purpose, or result [12]. For the aims of our analysis, synthetic data is not likely to relate to the natural person in content because it will most obviously not directly concern the individual as a result of the inclusion of randomness in the data generation process. This is, the content of synthetic data and the content of personal data would differ in substance, therefore breaking relatedness *stricto sensu*. However, where synthetic data presents data points which preserve the characteristics of the original data with high accuracy and/or statistical outliers are present, synthetic data could relate to the individual in content [19]. Besides this, synthetic data will relate to the natural person more obviously in purpose or result. It will relate to the natural person 'in purpose' where the data controller or a third party makes use of synthetic data with the goal of evaluating, treating in a certain way, or influencing the status or behavior of the data subject [12]. For instance, synthetic data can be used with the aim of evaluating the properties of a certain group without processing their personal data. In these cases, synthetic data would certainly relate in purpose, but it will not amount to personal data processing as previously argued. It will relate to the natural person 'in result' where synthetic data is likely to have an impact on the person's rights and interests. For instance, when the individual is treated differently from other people as a consequence of the processing of synthetic data. However, as previously argued, identifiability would need to be triggered to consider synthetic data as personal data in these cases.

In general, the case study of the 'relating to' element can be varied, and contextual elements will need to be taken into account for the legal qualification of synthetic data, thus increasing the complexity of this assessment compared to the previous elements.

### 5.2.2 'Identified or identifiable'

The most problematic validation is, however, that concerning the 'identified or identifiable' class, as it constitutes the open yardstick upon which the notion of risk most obviously comes about. An important point of the assessment of this element is the double standard of consideration that identifiability presents for European data protection law. As opposed to relatedness, which is assessed in confrontation with one single element, at least as extracted from the tenor of the law, identifiability in data protection law is specifically positivized as a double standard, *i.e.* the 'identified' and 'identifiable' classes. This categorical unfolding of the notion of identifiability enlarges the complexity of its assessment, as both direct and indirect methods are rendered consubstantial to the concept and, therefore, they both need to be equally considered.

**A**t its core, the 'identified or identifiable' element comprises an identification threshold constituted by two limits. The upper limit represents the natural person who has been 'identified', whereas the lower limit represents the natural person who has not been identified yet, but who is possible to be so [12]. A natural person can be 'identified' within the upper limit by reference to direct identifiers, such as the full name or address, or indirect identifiers, such as age, occupation, or place of residence [12]. At the same time, a natural person can also be 'identifiable' within the lower limit by reference to direct or indirect identifiers in combination with other pieces of information, including, but not limited to, gender, ethnic origin, or health status [12]. If the processing of any information relating to the natural person falls within the scope comprised by these two limits, it is presumed that said processing may constitute a risk to the fundamental rights and freedoms of individuals worth triggering European data protection law [15] and, consequently, the corresponding data protection obligations.

**I**n assessing the identifiability threshold, two main dimensions need to be considered: a subjective and an objective one. The consideration of these two dimensions is a natural consequence of the organic articulation of the European data protection framework, which is built on subjects (actors subject to data protection law) and objects (personal data). The subjective dimension, on the one side, refers to the standard of proof of actors that will be considered in the assessment of identifiability. The standard of proof is subject to 'either the controller or another person' [20]. While the 'controller' is a defined concept in the GDPR [21], 'another person' is a concept that does not fit well within the set of actors contemplated in the GDPR, *i.e.* controller [21], processor [22], recipient [23], third party [24], and data subject [25]. Notwithstanding this, a purposive interpretation of data protection law would lead the regulatee to understand that 'another person' refers to any of the data protection actors. The objective dimension, on the other side, refers to the standard of proof of factors that will be considered in the assessment of identifiability. The standard of proof for the evaluation of the factors is 'reasonable likeness', a concept which is left undetermined by the legislator [20]. According to the GDPR, these include costs, time, available technology, and future technology [20]. As will be discussed in the following, the assessment of these dimensions rises an exegetical problem on identifiability that confronts different approaches and authoritative and doctrinal positions.

**F**rom the viewpoint of the subjective dimension, the identifiability threshold is deprived of determinative clarity as to the standard of proof of actors to measure the risk of re-identification or, in other words, as to the amount of actors that should be considered by the processing entity in assessing whether a person can be identifiable or not. Given the indetermination of the standard of proof, two diametrically opposed lines of thought have emerged to fill in its content, better known as the 'absolute' and 'relative' approaches.

**O**n the one side, the absolute approach represents the viewpoint under which the consideration of actors is made on the basis of any theoretical probability of identification in relation to the processing of personal data. This kind of approach forces the processing entity to take into account in its risk assessment all *possible* actors that can re-identify the data subject. As such, the absolute approach constitutes the highest level of abstraction in the assimilation of actors for the assessment of the identifiability threshold. On the other side, the relative approach represents the viewpoint under which the consideration of actors is made on the basis of the realistic probability of identification in relation to the processing of personal data. This kind of approach forces the processing entity to take into account in its risk assessment all *probable* actors that can re-identify the data subject. As such, the relative approach constitutes a lower level of abstraction in the assimilation of actors for the assessment of the identifiability threshold. Thus, both approaches tend to form a dichotomy between theoretical and realistic considerations on identifiability. While the absolute approach focuses on the *possible* actors, the relative approach focuses on the *probable* actors.

**B**ecause processing entities tend to emphasize one of both approaches, the legal qualification of synthetic data is torn between these two limits. Depending on the adopted approach, different assessment scenarios can be guessed. If an entity is driven by the absolute approach, it may emphasize in its assessment of re-identification the broader boundary condition of 'by another person'. This would imply that the processing entity will be prone to consider any other actor subject to data protection law as a potential adversary capable of achieving re-identification. If, contrarily, a processing entity is driven by the relative approach, it may emphasize in its assessment of re-identification the narrower boundary condition of 'by the controller'. This would imply that the processing entity would consider itself, and eventually its closest counterparties, as the potential adversaries capable of achieving re-identification. As can be seen, both postures aim at realizing data protection goals. However, the way in which they respond to those objectives is of different degree.

Closely related to this point, we contend that the same subjective scope used for the assessment of identifiability should be used for the assessment of relatedness. This implies that identical approaches must be followed to establish whether any information relates to the natural person, and to whether any information makes the natural person identified or identifiable. If an entity follows, for example, a relative approach for the establishment of identifiability, it would not make sense to consider relatedness from an absolute perspective, as the subjective dimension of identifiability and relatedness is a shared attribute of both elements. Theoretically, it could be possible. But such possibility should not have legal consequences on the categorization of whether certain information is considered personal data or not if consistency is to be prioritized. The same holds true where identifiability is assessed from an absolute perspective. If the processing entity increases the legal standard for the assimilation of actors for identifiability to absoluteness, it cannot restrict relatedness to a relative approach, as it would cause conceptual inconsistency in the measurement of means. Henceforth, harmonization in the criteria used for assessing the elements of personal data should be a regulatory goal. Independent assessments and discretionary decisions in the treatment of both elements should be avoided if a harmonized framework is to be safeguarded and legal certainty is to be maximized.

From the viewpoint of the objective dimension, the identifiability threshold is deprived of determinative clarity as to the standard of proof of means to measure the risk of re-identification or, in other words, as to the extent to which objective factors should be considered in assessing whether a person can be identifiable or not. Given the indetermination of the standard of proof, or 'reasonable likeness', two risk assessment frameworks can be distinguished, the so-called 'zero-risk' approach [26] [27] and, in the absence of literature providing a name for the other one, we would call 'acceptable-risk' approach.

On the one side, the 'zero-risk' approach represents the viewpoint under which the consideration of means is made on the basis of any theoretical probability of identification in relation to the processing of personal data. By emphasizing this perspective, the 'zero-risk' approach seeks to theoretically reduce the risk of re-identification to zero. This kind of approach forces the processing entity to take into account in its risk assessment all possible objective factors that can re-identify the data subject. As such, the 'zero-risk' approach constitutes the strictest boundary condition for the evaluation of identifiability. On the other side, the 'acceptable-risk' approach represents the viewpoint under which the consideration of means is made on the basis of the realistic probability of identification in relation to the processing of personal data. By emphasizing this perspective, the 'acceptable-risk' approach seeks to theoretically reduce the risk of re-identification to an 'acceptable' level in which the re-identification of the data subject becomes minimized, but not eradicated. This kind of approach forces the processing entity to take into account in its risk assessment all probable means that can re-identify the data subject. As such, the 'acceptable-risk' approach constitutes a more relaxed boundary condition for the evaluation of identifiability. Thus, both approaches tend to form, again, a dichotomy between theoretical and realistic considerations on identifiability. While the 'zero-risk' approach focuses on the *possible* means for re-identification, the 'acceptable-risk' approach focuses on the *probable* means for re-identification.

Because processing entities tend to emphasize one of both approaches, the legal qualification of synthetic data is torn between these two limits. Depending on the adopted approach, different assessment scenarios can be guessed. If an entity is driven by the 'zero-risk' approach, it may emphasize in its assessment all possible means that could be used for re-identification, such as the database from which the synthetic data was generated, but also other related public databases and information, and beyond. It may consider, in addition, a wide variety of privacy metrics to assess identifiability and a wide range of re-identification technologies, in the present and the future, which may put at risk the identifiability of the synthetic dataset, through an extensive period of time. Finally, it may have a very strict standard for the amount of time and cost that would be needed to reidentify the data subject. If, contrarily, a processing entity is driven by the 'acceptable-risk' approach, it may consider in its assessment of re-identification the database from which the synthetic data was generated, and possibly some other related public information, if relevant. It may consider, in addition, the use of a limited number of privacy metrics to assess identifiability and a limited number of re-identification technologies, probably those that could be expected to be used in the present and the future, which may put at risk the identifiability of the synthetic dataset. It would also realistically set the time boundaries to a limited period in which it predicts that current technology would become obsolete, and it may have a practicable standard for the amount of time and cost that would be needed to reidentify the data subject by using specific means. As can be seen, both postures

aim at realizing data protection goals. However, the way in which they respond to those objectives is of different degree.

**D**isagreement still exists about the appropriateness of the exposed approaches, both at subjective and objective level, to construe of identifiability. While the authoritative understanding of these approaches seems to favour the absolute and zero-risk approach [28], technical scholarship advocates more and more for a relative and acceptable-risk approach [26]. Differences are also evident between the civil and common law traditions. While civil law systems seem to defend the authoritative interpretation of identifiability [29], common law systems seem to defend the technical interpretation [30]. As a result, the determination of the 'identified or identifiable' element and, consequently, of relatedness, still remains an unresolved issue which, in turn, impregnates the assessment of whether synthetic data is personal or not.

## 6   Synthetic data as anonymous data

Anonymous data is defined in Recital 26 GDPR as:

> *'information that does not relate to an identified or identifiable natural person or to personal data rendered anonymous in such a manner that the data subject is not or no longer identifiable.'*

**B**ased on this definition, synthetic data is being increasingly defended as an effective anonymisation technique to render personal data anonymous in such a manner that access, analysis, sharing, reuse, and publication of data can be carried out without revealing personal information. One must note, however, that the asseveration of synthetic data as anonymous comes with certain intricacies.

**I**n the first place, data synthesis is subject to a balancing test between utility and anonymity [4]. While utility can be understood as a measure of the satisfaction of synthetic data to produce analysis results similar to those that the original data would produce, anonymity should be understood in the same terms as described above. As a rule of thumb, the higher the utility of a synthetic data set, the lower its anonymity [4]. The approach adopted to achieve anonymity and privacy must be understood in the terms of the previous discussion on the subjective and objective dimensions of identifiability. If a synthetic data set maximises utility by fitting the original data set very carefully, anonymity would be lost because the synthetic data set would be a replication of the original data set. If a synthetic data set maximises anonymity by fitting the original data set very carelessly, the utility would be lost because the synthetic data set would be statistically different from the original data set. It is as relevant to optimise the utility of the synthetic data set as it is to prevent the re-identification of the natural person [31]. The trade-off between utility and anonymity must be, therefore, correctly navigated to generate appropriate synthetic data.

**A**t the same time, one must consider that the very nature of this trade-off is at odds with the plausibility of generating completely anonymous data sets, or data sets with zero risk of re-identification, if utility also needs to be preserved. This forces one to consider anonymity in the creation of synthetic data sets in probable terms, potentially resembling a preference to the relative and 'acceptable-risk' approaches. As a result, the determination of whether a synthetic data set complies with the required anonymity standards or not should be answered, inter alia, by considering the probability of re-identification that said synthetic data set has in relation to an acceptable threshold. If data synthesis is carried out poorly, the risk of re-identification can become higher, given the greater chance of record replication. On the contrary, if data synthesis is carried out properly, the risk of re-identification can be minimised. The probability of re-identification can be measured by using different metrics [32].

**B**ased on the previous assumptions, supporters of synthetic data argue that, where synthetic data is properly generated, there is no one-to-one mapping from synthetic records back to the person and therefore consider synthetic data as anonymous data [4]. Of course, such a premise should be considered in statistical terms, taking into account the above-mentioned utility-anonymity trade-off. This means that, where synthetic data is properly generated, it is, statistically speaking, indistinguishable from the original data such as to trigger the anonymisation standard. In these terms, synthetic data is argued to eliminate the risk of re-identification and provide for strong data protection guarantees. Opponents of synthetic data contend that even where it is properly generated, one-to-one

relationships are still possible, particularly if the synthetic data set preserves the characteristics of the original data set with high accuracy and/or statistical outliers are present [19]. Opponents make use of this argument to consider synthetic data as identifiable information. In contextualising the previous approaches with respect to our analysis on identifiability, one can distinguish that the question of whether synthetic data is considered anonymous or not is actually a problem concerning the introduced dimensions of the identifiability test. On the one side, supporters of synthetic data as anonymous data can be situated within the spectrum of the relative and 'acceptable-risk' approaches. On the other side, opponents of synthetic data as anonymous data can be situated within the spectrum of the absolute and 'zero-risk' approaches. As can be noticed, no consensus has been reached yet in the literature.

**I**n addition to the previous, one must also consider that the application of data synthesis would not circumvent on its own the European data protection framework. At its core, anonymisation encompasses not only a set of techniques, but also technical and organisational safeguards designed to prevent re-identification over time [20]. This is inferrable from the notion of risk inherent to the concept of personal data, which needs to be assessed contextually, as well as from the tenor of the law, which emphasises, as introduced in the explanation of the objective dimension of identifiability, the consideration of the objective factors in the assessment of anonymity [20]. According to Recital 26:

> '*[t]o ascertain whether means are reasonably likely to be used to identify the natural person, account should be taken of all objective factors, such as the costs of and the amount of time required for identification, taking into consideration the available technology at the time of the processing and technological developments.*'

**O**ne final concern of synthetic data is the possibility of inferring sensitive information about the individual yet still where the identifiability test does not render a positive result. This refers to the cases in which the natural person is not identified nor identifiable, but sensitive information can be still inferred from him or her. In technical jargon, this would equate to the risk of attribute disclosure [33], where one learns something about an individual from the data set with some level of certainty, independent of whether identification concurs or not. In these cases, the problem of data synthesis amounts to a problem of choice of the desirable regulatory model for data protection law: a model that prevents identifiability and/or a model that prevents information inference. As previously introduced, synthetic data aims to tackle data protection from an identifiability perspective or, in other words, it aims to ensure that the records of the individual would not be singled out or linked. If, however, an adversary knows of the presence of an individual in the original data set, even if that individual cannot be individualised, sensitive inferences might still be possible [28]. According to the opinion of Article 29 Working Party (A29WP), an institutional group with authoritative opinion in the field of European data protection law that was replaced by the current European Data Protection Board, to consider personal data as 'truly' anonymous, inferences about the characteristics of the individual must be ruled out [28]. While such a restrictive interpretation of anonymisation enhances the protection of personal data and, consequently, the protection of other fundamental rights and freedoms, it is in material disconnection with data protection law, which focuses on an identifiability substrate, as extracted from the definition of personal data in Article 4(1) GDPR and Recital 26 GDPR. In other words, while the risk of singling out an individual or disclosing its identity is easily assimilated by the identifiability threshold, the risk of inferring attributes of the person possesses a more difficult accommodation according to the tenor of the law. Following this line, there is a need to discuss the extent to which the recommendations of A29WP help model and enforce the interpretation of the data protection framework and, more generally, the data protection model that society deems adequate.

## 7   Synthetic data as pseudonymous data

According to Article 4(5) GDPR, pseudonymous data is:

> '*personal data that cannot be attributed to a specific data subject without the use of additional information, provided that such additional information is kept separately and is subject to technical and organisational measures to ensure that the personal data are not attributed to an identified or identifiable natural person.*'

While the GDPR does not define the concept of attribution, we understand that the concept refers here to the use of additional information that would make the data subject identifiable. If properly generated, synthetic data cannot be attributed to a specific data subject, given its eugenic nature. This means that the use of additional information may not pinpoint the data subject, therefore circumventing the identifiability test. Nonetheless, synthetic data can still show sufficient structural equivalence with the original dataset or share essential properties or patterns to trigger attribution [19]. For instance, if the synthetic data is generated by one-to-one transformation of the original dataset so that each synthetic datapoint equates to an original data point, source features would be substantially maintained in the synthetic data set and hence could fall under the definition of pseudonymous data. This might be the case where the trade-off of data synthesis is not properly navigated, and the original data set is kept by the controller and used as additional information to draw personal attribution. In such cases, the data protection obligations will apply *tout court.* One must note, however, that such a conclusion will be determined, in the first place, on the desired approach to identifiability, thus rendering different results based on the chosen data protection model.

Closely related to this, it would be possible that, in certain cases, synthetic data might be purposely generated by the data controller with the intention of treating it as pseudonymised data. This might be the case where the utility of the data is considered important, at the expense of anonymity. For these specific contexts, although synthetic data would fall under this category, data synthesis would notwithstanding be considered a useful privacy by design measure for protecting the rights and freedoms of individuals as well as one compliant with the data minimization principle. This can allow researchers, in certain circumstances, to benefit from the advantages of data synthesis while ensuring compliance with data protection law.

## 8    Conclusion

Where data synthesis is carried out from original (personal) data, as analysed in this paper, compliance with the European data protection legislation would be necessary, at least in the phases prior to data synthesis. This suggests that the controller would still need to have a lawful basis to collect personal data and be subject to the corresponding data protection obligations in relation to the type and sensitivity of the collected data and the aims pursued. Only after personal data has been rendered synthetic in such a manner that the data subject is no longer identifiable, synthetic data will be considered anonymous. Yet, one must note that the bar of anonymisation has been set very high by the European legislator. It may comprise anonymisation techniques, such as data synthesis, and post-anonymisation control mechanisms, both technical and organisational. In this sense, the question of whether synthetic data remains anonymous is not a discrete but a continuous issue. It depends on the extent to which the synthetic data deviates sufficiently from the original data to avoid identifiability and the extent to which anonymity is sustained over time. To validate the former, a formal assurance of identifiability must be performed by the controller on the dataset after the data synthesis to validate whether re-identification is possible. To validate the latter, technical measures, such as confidentiality, integrity, availability, and resilience measures, as well as organisational measures, such as security management, incident response, and business continuity, human resources, and test, assessment, and evaluation measures, must be in place [34]. Yet still, the possibility of deducing, with significant probability, attribute values from synthetic datasets remains an unresolved issue. In the same line, the question of how to categorize synthetic data with respect to the identifiability test and, consequently, to relatedness, poses the challenge of creating societal consensus on which of the approaches, both at the subjective and objective level, are deemed to be more adequate. For this reason, any attempt to legally qualify synthetic data with academic rigour would necessarily first have to resolve the above question.

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
