# OpenReview forum: "On the legal nature of synthetic data"
_NeurIPS.cc/2022/Workshop/SyntheticData4ML — Neurips 2022 SyntheticData4ML_

### Official Review · Reviewer_QeFS · 2022-10-17
**We need both lawyers and data scientists to think through the legal and regulatory implications of synthetic data.**

**Rating:** 7
**Confidence:** 4

**Review:**

The ML community needs lawyers to think through the legal and regulatory implications of synthetic data. The most important feature of this paper is that it apparently has been written by lawyers. I say 'apparenty' because with blind review I can't say for sure what the qualifications of the authors are. I had to look up the term 'casuistry'; it turns out the meaning of this word depends on whether you are a lawyer or not:
* _Legal definition_: a process of reasoning that seeks to resolve moral problems by extracting or extending theoretical rules from a particular case, and reapplying those rules to new instances.
* _General English definition_: use of clever but unsound reasoning (Oxford dictionary)

We need people who can apply the first defintion, not the second!

This topic involves many technical issues. For example, even the discussion of what "zero risk" means has a technical side. Imagine a data synthesizing process to invent names for simulated people. It could use a database of real names to find statistical distributions of given names and surnames, then randomly choose samples from these distributions to create new combinations. The probability of this random process producing combinations of given name and surname that actually exist in the dataset is not zero, and if it also generated random numbers for height, weight, and blood pressure, these would likewise have a non-zero probability of matching a real person. But the process gives absolutely zero clue as to which data records happen to match reality. So this is zero risk in one sense, but not in another.

To address these issues, or even to explore them in depth, require both technical and legal expertise. Having interested lawyers at the SyntheticData4ML workshop could be a useful step in this direction. One could imagine fruitful conversations about differential privacy budgets inter alia.

---

### Official Review · Reviewer_EeH9 · 2022-10-19
**Interesting work on legal analysis of synthetic data**

**Rating:** 7
**Confidence:** 4

**Review:**

A legal analysis on synthetic data's privacy preservation property is much needed for the ML community. Although I do not have the required legal knowledge to evaluate the technical correctness of this work, I believe that this work represents an early but important step towards a better understanding of the legal status of synthetic data.

An area of future work is to study the various privacy-preserving generative models, such as the ones using differential privacy [1, 2, 3] and other notions of privacy [4].

[1] Ho, Stella, et al. "DP-GAN: Differentially private consecutive data publishing using generative adversarial nets." Journal of Network and Computer Applications 185 (2021): 103066.

[2] Jordon, James, Jinsung Yoon, and Mihaela Van Der Schaar. "PATE-GAN: Generating synthetic data with differential privacy guarantees." International conference on learning representations. 2018.

[3] Zhang, Jun, et al. "Privbayes: Private data release via bayesian networks." ACM Transactions on Database Systems (TODS) 42.4 (2017): 1-41.

[4] Yoon, Jinsung, Lydia N. Drumright, and Mihaela Van Der Schaar. "Anonymization through data synthesis using generative adversarial networks (ads-gan)." IEEE journal of biomedical and health informatics 24.8 (2020): 2378-2388.

---

### Meta-Review · Area_Chair_mpXa · 2022-10-19

**Recommendation:** Accept